# COVID-19 and the Fears of Italian Senior Citizens

**DOI:** 10.3390/ijerph17103572

**Published:** 2020-05-20

**Authors:** Diego de Leo, Marco Trabucchi

**Affiliations:** 1Italian Psychogeriatric Association, 25121 Brescia, Italy; trabucchi.m@grg-bs.it; 2De Leo Fund, 3139 Padua, Italy; 3Australian Institute for Suicide Research and Prevention, Griffith University, Mt Gravatt campus, Brisbane 4122, Australia; 4Department of Pharmacology, Tor Vergata University, 00133 Rome, Italy

**Keywords:** SARS-CoV-2, COVID-19, older adults, nursing homes, senior dwellers

## Abstract

Italy has been hit very hard by the severe acute respiratory syndrome—coronavirus-2 (SARS-CoV-2) pandemic. This brief report highlights some of the peculiarities manifested by its older adult population, with particular reference to those living in nursing institutions and at home. Mortality data (as of 26 April) are reported, together with reactions to forced isolation, loneliness, and fear of contracting the disease, which represent big challenges for all, especially for frail elderly people.

## 1. COVID-19 in Italy

In Italy, the severe acute respiratory syndrome—coronavirus-2 (SARS-CoV-2) developed with extreme virulence, presenting by March an official number of infected and deceased individuals higher than that of China, where the pandemic first appeared. As of 26 April, the United States of America has the largest number of infected people, followed by Spain, whose infected count now exceeds the number of infected people in Italy by a few thousand. It is worth mentioning that these figures are based only on people who tested positive to the swab and do not reflect the real impact of the infection on the general population, which remains unknown. Furthermore, testing modalities and cultural factors could explain at least some of the observed differences between countries. However, Italy is the second country—after the U.S.—in terms of mortality, with 26,977 deaths [1]. The vast majority of these deaths come from individuals aged 70+ years (more than 84%), with approximately 9500 deceased individuals aged 83+ years [2]. 

For more than fifty days, coronavirus disease 2019 (COVID-19), caused by the SARS-CoV-2, has forced 60 million Italians into their homes. Schools are closed; only 30% of all commercial and industrial activities are functioning. The pandemic is severely testing the entire Italian system, in particular its welfare structures. The northern regions were the hardest hit, with Lombardy, Emilia-Romagna, Piedmont, and Veneto leading the ranking. To cope with the emergency, intensive care units were built in record time and entire hospitals were converted into COVID-19 care. However, the pandemic found the country largely unprepared, failing to provide on time the necessary protection even to its health workers, with serious shortcomings regarding the supply of eyeglasses, masks, gloves, and gowns. Many health professionals lacked the necessary disaster preparedness. So far, this has resulted in the loss of 151 doctors and 39 nurses [2]. Difficulties in assisting critical situations, combined with the scarcity of places suitable for the reception of patients in serious conditions and lack of a sufficient number of ventilators, have given rise to very painful ethical choices for health professionals on how to prioritize patients for care with the available equipment [3]. 

## 2. Senior Citizens and the Pandemic

The authors of this report are the vice-president (DDL) and the president (MT), respectively, of the Italian Psychogeriatric Association. During the pandemic, they maintained close contact with many colleagues, mostly geriatricians, neurologists, and general practitioners (in total 52 physicians), operating in different contexts in Italy—hospitals, nursing homes, and their patients’ homes. This article is not written in the form of an original research article, but it is intended to be a commentary of what is happening in Italy. The contents of this report condense the narratives that describe the experiences of those front-line medical doctors with their senior patients during the tragic spread of SARS-CoV-2. The original narratives of 48 of those physicians were included in a special issue of the journal of the Italian Psychogeriatric Association [4].

Older adults with chronic poly-pathologies were and still are the subjects who have suffered the most from the reaction difficulties of the Italian health system. Before the pandemic, even those living in senior housing communities (designed to reduce social isolation) reported significant levels of loneliness [5]; undoubtedly, these are now exacerbated by the rules imposed in quarantine and social isolation. In a Journal of American Medical Association *Viewpoint*, Steinman et al. [6] eloquently described the general condition of older adults isolated at home as a consequence of the pandemic. Here, we intend to add some considerations on the peculiar conditions suffered by Italian elders.

In Italy, in cities of Lombardy such as Bergamo and Brescia, the deaths occurred at an impressive rate, giving rise to the sad televisions images of long lines of military trucks carrying coffins to incinerators, often very far from the place of origin of the deceased. There was no way to celebrate funeral rites, nor burial in the cemeteries of the place of residence. One of the most heartbreaking aspects of the loss of life associated with this pandemic was the inability to accompany loved ones in their last moments of life. The isolation imposed by the pandemic means that thousands of subsequently deceased subjects were last seen when an ambulance took them to be hospitalized.

Nursing homes, neglected for too long by government administrations, have paid a very high price due to the lack of protective measures and the social distancing that COVID-19 has imposed. In the province of Bergamo (Lombardy), in just twenty days—from 7 March to 26 March—over 600 deaths occurred in nursing homes with a total accommodation capacity of 6400 beds [7]. To date, it has been calculated that almost a quarter of all residents in nursing homes in northern Italy have died due to COVID-19 and that the average age of Italians who lost their life with the virus was 79 years [2].

Older people have been specifically advised to stay home given their particular vulnerability to COVID-19, but also to reduce the burden on health services and limit the spread of the disease. The negative effects of isolation could be particularly insidious for older adults and people with pre-existing mental illness [8]. Regular checks by doctors and regular intake of drug therapies can become problematic. Alcohol intake and smoking can increase frequently. Eating properly and maintaining personal hygiene can become quite difficult. These aspects can increase the sense of demoralization and despair in people forced to live alone and easily suffering from loneliness and social isolation, well-known risk factors for suicide in late life [9]. Indeed, self-killing cases have been repeatedly reported in the national press [10] and can be expected as one of the most feared outcomes of the pandemic [11].

## 3. Nursing Homes as Besieged Castles

Nursing homes have become like castles under siege, where old guests can no longer leave and new guests can no longer enter, given the spread of the infection within these institutions [6]. Healthcare professionals who work there report some peculiar elements of the behavior of the guests of those institutions. Residents face fear of the disease and anguish for its threatening consequences with mixed attitudes. These attitudes range from continuous prayer (a rosary to pray has become a frequent request from residents) to a nihilistic form of fatalism (“There is nothing I can do; I can only hope death will come without too much suffering”). Episodes of psychomotor agitation that require drug sedation are also quite frequent. Residents are often seen crying constantly: they have lost their roommate or people they have made friends with in the nursing home. Health workers wearing protections frighten them, because this reminds them that the virus is highly contagious. Therefore, getting infected and having the symptoms of the disease would just be a matter of hours.

When health conditions actually deteriorate, many residents express the wish of avoiding hospitalization. This is because they do not believe in a favorable treatment outcome and because they think they would only encounter unnecessary pain in complete detachment from the surrounding world. They believe that there are no effective therapies and that none of the protocols operating in hospitals would have a positive outcome. Given that residents usually watch television for most of the time, they are now aware that the vast majority of deaths are concentrated among elderly people, especially those who suffer from chronic conditions (as they know they are). If they were to get COVID-19, they do not want to be intubated but hope to be treated by someone they already know (possibly the same doctors of the nursing home) and, overall, to have the possibility to stay in touch with family members and loved ones [12]. By this, they mean “physical” contact; in fact, many reject the tablet offered for video communications by health workers. 

## 4. The Elderly and the War Data Bulletin

For people living in their homes, the situation is not very different from that in rest institutions. Italian elders have become accustomed to days sadly marked by the 6.00 p.m. television bulletin of the Civil Protection, which communicates the number of infected people and—above all—number of people who died. In the eyes of older adults of Italy, those bulletins never miss the opportunity to point out that the greatest number of deaths occurs in the most-advanced age groups, with a lethality index (number of subjects positive to swab tests/number of deceased subjects) that reaches 30% among those who are 80–89 years old, the group that comprises most residents of Italian nursing homes.

Once again, according to data from Istituto Superiore di Sanità, 60.7% of deceased subjects suffered from at least three pre-existing pathologies (e.g., cardiovascular disorders, diabetes, kidney problems, etc.). They died, on average, ten days after the clinical onset of symptoms (especially fever, dyspnea, cough, etc.) and five days after hospitalization, which in turn occurred five days after the onset of symptoms [2].

Senior citizens living with family members often refuse to wear mask and gloves at home, although they are told that the virus could spread easily among cohabiting people, especially in small apartments. It would be the “will of God”, they often say. Similarly to the case of nursing home residents, they say, “If I get sick, do not take me to the hospital; treatments are useless there and, in any case, I’m too old. So, it would be OK for me to pass away; I just want to die in my bed”. 

These days, many elders are dictating their last will. While the say to their children not to worry too much for them, they often ask for the presence of a doctor certifying that they are “compos mentis”. In this way, their will would have legal validity. Another concern is the fear of being buried in mass graves. These elders are aware that nobody can hold a funeral in present times; considering this, they hope to be cremated, with their ashes arranged in an identifiable cinerary urn. Most elderly people live alone; many have limited or no Internet knowledge and are poorly connected with other family or community members [13]. Quite frequently, some elderly people die in complete isolation; their corpses can be discovered many days after death [14]. 

As said, older adults did not grow up in an Internet-connected environment. Using a laptop or a tablet can be a challenge. To call their children and grandchildren—if ever in their lives—most of them still use an old telephone, not a smartphone. Often, television is the only company they have, as well as their most efficient time killer. “Can I survive it?” is the big question coming from them. The more time passes, the more likely they are to survive, they think. However, the long wait is unnerving.

Especially in industrious regions of northern Italy, the unusual silence that looms in cities contributes to making the atmosphere particularly surreal. Many elders wonder why this pandemic arose: what caused it? Some elderly people think that COVID-19 is the consequence of divine punishment for a society that has lost any moral value. Despite being completely isolated from the rest of the world, other elders do not feel safe inside their homes: they believe that the virus is in the air; it could enter through the windows or from any possible crack or gap in the walls of the house. Hence, contagion would be inevitable and only a matter of time. In these cases, anxiety levels are at the top, and ringing the general practitioner (GP) becomes an urgent need. If the GP does not answer, the next calls are to the police or the emergency services or to one of the numerous helplines activated in recent times to meet people’s psychological needs. 

## 5. Conclusions

In Italy, COVID-19 has violently struck a country that was insufficiently prepared to face it. The pandemic has overcharged a healthcare system (particularly its acute care wards) and confined millions of citizens at home or in nursing homes. Especially in northern Italy, healthcare professionals were overworked and operated on the edge of their forces and energies and fatigue. Poor disaster preparedness and precarious availability of protective equipment also contributed to the level of stress of doctors and nurses. It seems quite obvious to emphasize that adequate responses to dealing with disasters would require both specific training and appropriate equipment: these appear as mandatory recommendations for future challenges.

Like never before, people’s mental health must be supported in every possible way. Forced isolation, loneliness, and fear of contracting the disease are tough challenges for all people, especially for frail, elderly people [8]. In the current situation, promoting self-help and positive coping and reducing isolation also through technology appear imperative. Active outreach seems a necessity, especially for older adults [15]. This would help to counteract the feelings of abandonment and disempowerment that COVID-19 is imposing on all members of the community, with older adults being the most exposed ones.

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
