# Peer review of "COVID-19 and the Fears of Italian Senior Citizens"

_ijerph, 2020, doi:10.3390/ijerph17103572_

Round 1

Reviewer 1 Report

Thank you for this timely report on the situation in Italy. Although publishing this information as soon as possible is essential, it seems imperative to highlight the ethical aspects of the overcharge of the healthcare system (i.e., acute care), the confinement of citizens in nursing homes or at home and the ethical challenges that healthcare professionals are confronted with regarding overcharge, being at the limit of their forces and energies and fatigue. At the very least, these issues should be mentioned in this manuscript, e.g., in the conclusion section. It is also recommended to point out that healthcare professionals need to be better prepared - not only regarding materials - but also in terms of "disaster preparedness". Such a recommendation could also be introduced in the conclusion section. 

Please note that most probably the numbers purported by the Johns Hopkins Whitings School of Engineering, Mapping 2019-nCov (https://systems.jhu.edu/research/public-health/ncov/) may be accurate, but are dependent on testing modality and culture in the respective countries. Hence, the number - even if official - are to be consulted with some caution. 

Sincerely, 

Your reviewer. 

Author Response

Dear Reviewer,

Thank you very much for your useful comments, which I accepted in full and implemented in the manuscript. Accordingly, I have changed completely the first part of the Conclusion section.

I have also specified that figures are dependent on number of people tested, and modalities of testing and country culture. Very useful indeed. Thank you again for that.

Kind regards

Reviewer 2 Report

Clearly, the Covid-19 pandemic is a major health crisis with an unprecedented toll on human life and resources.

However, as much as this paper describes a very grave situation in Italy, one of the countries worst hit by the disease, it presents low scientific soundness as the data collection methods and the results are poorly described.

Data collection methodology are the "narratives describing the experiences of those font-line heath professionals with their senior patients during the tragic SARS-CoV-2 diffusion". There is no mention of the number of health professionals included in the report, nor their capacities, or their institution of employment, or any description of their characteristics (age, sex, duration of employment etc). There is no data collection guide to lead the reader through.

Consequently the results are presented in a hap-hazard way without any link to the data they came from. There is no information on the number of elderly people living in nursing homes in the areas presented in the report, nor any explanations on the statement "neglected by the government administrations for too long".

There is also mention of helplines set to assist people with their psychological needs but no information is given on their uptake or effect on the beneficiaries.

There is no mention of potential selection bias in the professionals included or the areas/nursing homes/houses covered in this report.

The effects of the pandemic on the mental health of the senior citizens is indeed a very important issue, but  this report presents some aspects more with a lyrical and not a scientific tone. 

Last, but not least, some English language check is required, as there is need for numerous expression and grammar corrections.

Author Response

Dear Reviewer,

Our contribution was not submitted as an original research article but as a Commentary. As such, we never meant to organise the script with the specific sections (e.g., Intro, Aims, Methods, Results, Discussion, Limitations, Conclusion) of an usual research paper.

As your comments implied, it is a 'tale', or - as we prefer - a report from the trenches.

Hence, many of the points you raised cannot be scientifically answered: we probably know the number of doctors, their work setting and their approximate age (not the number of years in service), but this was not the aim of this Commentary. The same holds true for number of helplines and their characteristics.

Your invitation to revise the English language was fully accepted and the paper went through a mother-tongue professional: The result was many corrections! (now in red in the manuscript).

Thank you very much for your attention on our manuscript.

Kind regards

Reviewer 3 Report

This brief report by Leo et al. focused on the mental health in patents with COVID-19. This is an important problem to be resolved beyond the treatment of the disease itself, especially in older adults. It seems timely. I agree with all of this manuscript. Authors may want to resolve several minor issues as below.

Major comments;

1) No major comment.

Minor comments;

1) Abbreviated disease and virus names of “COVID-19” and ”SARS-CoV-2” should be described with full terms as “coronavirus disease 2019” and “severe acute respiratory syndrome coronavirus 2”, respectively, at the first time of use in main text.

2) Then, please explain first that COVID-19 is caused by SARS-CoV-2 or SARS-CoV-2 is an etiological virus of COVID-19.

Author Response

Dear Reviewer,

Thank you very much for your precious comments, which are now fully integrated in our Commentary (see corrections in red into the manuscript).

We are grateful for your attention and work.

Kind regards

Round 2

Reviewer 2 Report

This revised version of the paper reads much better than the previous one; however the data collection methodology remains vague and the conclusions are rather general not fully supported by the results. Front-line professionals and senior patients where? At the nursing homes, at the hospital? How many professionals participated? Was there a structured interview?

The lack of disaster preparedness of the healthcare workers is blended with the adverse effect of the isolation measures on the mental health of the elderly population. is the level of disaster preparedness a fear of the elderly citizens as stated in the title? Please make the connection between the two clear.

Some further English editing is required especially in sections 3 and 4.

Once these points are addressed, the paper will be more specific in the message it tries to convey. 

Author Response

Dear Reviewer 2,

Thank you very much for your right and helpful comments to our report: we believe they have much improved its quality.

The publication online of a special issue of Psicogeriatria, the official journal of the Italian Psychogeriatric Association, permits now to have appropriate reference to the narratives of the physicians on which our article is based. This special issue refers to 48 out of 52 physicians that shared with us their stories and experiences during the pandemic. Four GPs did not publish their stories in the special issue, but their verbatim were precious to obtain info on dwelling elderly. 

Hopefully, we clarified the connection between preparedness and fears: Italian elderly seemed to be very skeptical of any form of therapy and felt very fragile and exposed to the risks of the virus.

We have obtained a further revision of the English language, which once more resulted as quite substantial (all corrections are in red).

Thank you again for your time on our short report.

With gratitude,

Diego de Leo & Marco Trabucchi